# In-Forest Planting of High-Value Herb *Sarcandra glabra* Enhances Soil Carbon Storage without Affecting the Diversity of the Arbuscular Mycorrhiza Fungal Community and Composition of *Cunninghamia lanceolata*

**DOI:** 10.3390/microorganisms10091844

**Published:** 2022-09-15

**Authors:** Hanchang Zhou, Tianlin Ouyang, Liting Liu, Shiqi Xia, Quanquan Jia

**Affiliations:** 1Jiangxi Academy of Forestry, Nanchang 330013, China; 2Jiangxi Provincial Forestry Science and Technology Experiment Center, Xinfeng 341600, China

**Keywords:** arbuscular mycorrhiza fungal, in-forest planting, *Sarcandra glabra*, soil nutrients

## Abstract

*Sarcandra glabra* in-forest planting, an anthropogenic activity that may introduce a variety of disturbances into the forest, is being popularly promoted in southern China, while its consequential influences on soil nutrients, as well as the arbuscular mycorrhiza fungal (AMF) community of key forest keystone plants, are still unelucidated, which hampers the assessment of ecological safety and the improvement of agronomic measurements. In this research, topsoil from a 3-year-old *Sarcandra glabra* planted forest and a nearby control forest were sampled, and the annual variation in the soil nutrients and AMF community of the keystone tree *Cunninghamia lanceolata* were investigated. Our result showed that the total amount of soil organic carbon of the *Sarcandra glabra* cultivation group was significantly higher than that of the control group (*p* < 0.05), which indicated that *Sarcandra glabra* cultivation significantly enhanced the topsoil carbon storage. Yet, there were only insignificant differences in the Shannon index and Chao index of the AMF community between the two groups (*p* > 0.05). PCoA analysis found that the compositional differences between two groups were also insignificant. This indicated that *Sarcandra glabra* cultivation had no significant influence on the diversity and composition of the *Cunninghamia lanceolata* AMF community. However, we found that the differences in the total amounts of nitrogen and total phosphorus between the two groups were relatively lower in April and September, which indicated the higher nutrient demands and consumption of *Sarcandra glabra* in these two periods and suggested that a sufficient fertilizer application in these two stages would reduce the potential competition for nutrients between *Sarcandra glabra* and *Cunninghamia lanceolata* in order to ensure *Sarcandra glabra* production and forest health. Lastly, our results reported a total extra income ranging from of CNY 127,700 hm^−2^ (7 years of cultivation) to CNY 215,300 hm^−2^ (10 years cultivation) provided by *Sarcandra glabra* in-forest planting, which indicated its powerful potential for mitigating poverty. Our research systematically investigated the annual variation in the soil nutrient content and keystone tree AMF community caused by *Sarcandra glabra* cultivation and offers constructive guidance for *Sarcandra glabra* cultivation and fertilization management and ecological safety assessment.

## 1. Introduction

Agroforestry is one of the economy-boosting strategies proposed by the United Nations, which has offered immense and positive contributions to several fields [1], including the resolving of regional poverty problems [1,2], maintaining of global food security [3] and mitigation of global climate change [4,5]. To date, agroforestry farming systems are closely related to the life of an estimated 1.2 billion people all over the world [3]. In-forest planting and non-timber forest-based products constitute a rising field of agroforestry, which offers a considerable house-hold income for several families, especially those in undeveloped regions [6,7]. However, in-forest planting may have negative effects on the forest, and several researches reported that agroforestry leads to decreasing biodiversity and threatens forest health [8]. For example, fertilization during herb cultivation can lead to soil eutrophication, lower ground water quality [9], and reduced soil microbial community diversity [10], and nutrient competition between crop and keystone trees might diminish the dominance of keystone plants and deteriorate the health of the ecosystem and agroforestry sustainability [11]. Hence, nearly all countries across the world are trying to find sustainable in-forest planting systems that are both economically and ecologically compatible with their local backgrounds.

Southern China has rich forest resources, which offer favorable conditions for the expansion of regional agroforestry economics [12] and especially favors the in-forest planting of high-quality herbs [12]. *Sarcandra glabra* is a Chinese traditional herb which has been used as medicine since nearly three thousand years ago [13]. As *Sarcandra glabra* is widely used in an expanding array of fields producing food, medical, and cosmetic products in recent decades, its market demand is sharply upsurging in China [14]. Moreover, the agroforestry cultivation of *Sarcandra glabra* is being proposed in southern China, though few assessments of the following influences on forest soils and keystone plants have been conducted and despite the fact that a series of extra anthropogenic disturbances have been introduced into the forest, such as fertilization, grass removal, etc. [13,14]. This is a potential but serious threat to ecological safety and relative industry sustainability.

The arbuscular mycorrhiza fungal (AMF) community has been reported to be an indicator of forest health, since the variation in the AMF diversity and composition are always accompanied by changes in the host physiological and ecological states [15,16]. The AMF community plays unique roles in maintaining the primary production and dominance of keystone species in coniferous forests [16,17], where *Sarcandra glabra* are usually planted. The AMF can not only help host plant, such as spruce (*Cunninghamia lanceolata*), to assimilate soil nitrogen and phosphorus via the hyphae [16,18], but also protects the host root from pathogenic microbe infection [17] and thus enhances the host plant’s dominance. Meanwhile, as the bridge linking the host plants to the soil, AMF communities have also been reported to be relatively sensitive to the changes in the soil nutrient content and stoichiometry, and the variation in the AMF diversity and composition can, in turn, influence the forest keystone plant’s state and scale, even affecting the whole ecosystem [15,18]. Hence, fertilizer application during the cultivation of herbs, such as *Sarcandra glabra*, may cause potential changes in the AMF diversity and composition, and further threaten forest health and regional ecological safety. However, studies of the effects of *Sarcandra glabra* in-forest planting on the AMF of forest keystone plants are still scarce, which deeply hampers the ecological safety assessment and the improvement of the agronomic measurement of *Sarcandra glabra* cultivation in needle forests.

To fill this gap, in this research, we selected a 3-year-old *Sarcandra glabra* planted needle forest and another nearby needle forest in order to investigate the annual variation in the soil nutrient and AMF communities of the keystone tree species and determined the influences of *Sarcandra glabra* in-forest planting on them.

## 2. Materials and Methods

### 2.1. Sites

The sampling site is located in Zhaokeng forest of Quannan county, Jiangxi Province, in southern China (114°42′3″ E,24°35′18″ N), and it has a subtropical monsoon climate, mean annual temperature 18.8 °C, and mean annual precipitation 1654 mm. The slope is 10~25° and south oriented. The keystone plant is *Cunninghamia lanceolata*, which was planted about 25 years ago (Figure 1). The density is about 1500 hm^−2^, the average diameter at breast-height is about 15.7 cm, and the coverage is about 75%. The main grass and shrub species are *Dicranopteris dichotoma* Berhn, *Spatholobus suberectus* Dunn, *Fissistigma oldhamii* Merr, *Ilex pubescens* Hook, *Callicarpa kwangtungensis* Chun, and *Carex breviculmis*. The *Sarcandra glabra* was planted in February 2017, and its density is about 30,000 hm^−2^. We used 10~15 cm branches of *Sarcandra glabra* for the vegetative reproduction. The bottoms of the branches were rinsed in a solution of ABT3 for 2~3 min, and then the branches were buried in 10 cm-deep soil at 5~10 cm intervals. The grasses and dwarf shrubs in the *Sarcandra glabra*-planted area (planted group) were removed artificially twice a year, and about 84 kg-N hm^−2^, 36 kg-P hm^−2^, and 70 kg-K hm^−2^ were applied in both April and September after the sampling [13]. In contrast, the forest nearby (the control group) was not planted with *Sarcandra glabra*, and it was subject to no relative anthropogenic disturbances.

### 2.2. Sampling

The topsoil (0~20 cm) was sampled in January (Jan), April (Apr), July, September (Sep), and November (Nov) 2020. Three subsample plots (20 m × 20 m) were randomly established in both the planted and control sites, and each one was located at least 20 m away from the others. A total of 20 soil cores (with an 80 mm diameter) were randomly chosen from the subsample plots, and each soil core was placed at a distance of at least 2 m from the other ones. The soil cores were mixed and passed through a 2 mm sieve to remove the litters and rocks. Then, about 500 g of the mixed soil sample was stored in −80 °C for further analysis. Root samples of the *Cunninghamia lanceolata* were dug up by shovel from the surface soils (0–20 cm) at 10 randomly selected locations in each subsample plot. The 10 sampling locations were located at least 2 m apart. We then used these roots and selected the distal first and second branch orders of the live *Cunninghamia lanceolata* roots, which are considered as absorptive roots [19], for further analyses. The identification of the live roots of the *Cunninghamia lanceolata* was based on root color, morphology, and the elasticity of their tissues. Following collection, all the absorptive root samples were gently washed with tap water and then deionized water to ensure that they were free from soil. We selected ca. 50 absorptive root segments for DNA extraction [20]. The seeds, labor and fertilization costs, yield, and price of the *Sarcandra glabra* were investigated at the local level.

### 2.3. Measurement of the Soil Properties

We measured the soil pH using a pH meter (FE20-FiveEasyTM pH, MettlerToledo, Giessen, Germany) after 30 min of shaking in a soil:water (1:2.5 *w*/*v*) suspension [21]. We measured the total organic carbon (TOC) and total nitrogen (TN) contents using an elemental analyzer (VarioMAX, Elementar, Langenselbold, Germany) [21]. The alkali fusion-Mo-Sb anti-spectrophotometric method was used to measure the total phosphorus content (TP) [21]. The alkali-dissolved flame-spectrophotometry method was used to test the total potassium (TK) [22]. We used the alkaline hydrolysis diffusion method to measure the alkali-hydrolyzable nitrogen (AN) content [21]. The available phosphorus (AP) was measured using the HCl-H_2_SO_4_-acid-dissolved Mo-Sb anti-spectrophotometric method [21]. We used ammonium acetate dissolved flame-spectrophotometry to test the available potassium (AK) [22]. Three repetitions were conducted to measure the soil properties of each sample, which corresponded to one molecular sequencing of that sample.

### 2.4. DNA Extraction and Sequencing

The DNA was extracted using a E.Z.N.A^®^ soil Kit according to the manufacturing protocol. The primer for the amplicon generation was the AMF-specific LR1-FRL4 [20]. Sequencing libraries were created using the NEBNext^®^ Ultra™ II DNA Library Prep Kit for Illumina^®^ (New England Biolabs, Ipswich, MA, USA). The Illumina Miseq PE300 platform was used for the sequencing and yielded 250 bp paired-end reads (Magigene Biotechnology Co., Ltd., Shanghai, China). The Magichand Cloud Platform (http://www.magichand.online, accessed on 4 April 2021) was adopted to conduct all the sequence bioinformatic scripts [23]. Sequences were trimmed to remove the barcodes and developed, with a minimum overlap length of 20 bp, into full-length sequences by FLASH after removing the barcodes and primers, in order to create pair-ended sequences [24]. We removed the chimeras using UPARSE, and the sequences that had a 97% or higher level of similarity were clustered into operational taxonomic units (OTUs) [25]. The Shannon index and Chao index were calculated using an online platform (http://www.magichand.online, accessed on 4 April 2021) according to the read number of each OTUs for every sample. A higher Chao index indicates that a sample contains a higher quantity of none-zero reads OTUs. A higher Shannon index indicates that a sample has a diverse OTU composition, with not only more none-zero reads but also more evenly distributed reads [25]. The AMF OTUs were annotated using the MaarjAM database [26].

### 2.5. Statistics

The principal coordinates analysis (PCoA), based on the Bray–Curtis distances at the out level, was conducted using the online platform (http://www.magichand.online, accessed on 4 April 2021) with R base programmes [23]. ANOISM (analysis of similarities) tests were used to define the significance of the differences between the control and planted groups, and a P of less than 0.05 indicated significance differences. Significant differences in the basic soil properties and diversity indices between the planted and control group were checked using *t*-tests in SPSS version 26, and a P of less than 0.05 indicated significance differences. Moreover, Pearson analysis was conducted to assess the basic soil properties and relative abundances of the top 20 AMF OTUs in SPSS version 26, and a P of less than 0.05 indicated a significant correlation. The figures were generated using Origin 2019.

## 3. Results

### 3.1. The Variation in the Soil Nutrients

The annual variation in the basic soil nutrients of both the planted and control groups is depicted in Figure 2. The pH ranged from 4.34 (planted, Nov) to 4.57 (control, Apr and Sep), and the pH in the planted group was generally lower than that at the control (*p* < 0.05), especially in Nov (about 0.16 lower) (Figure 2A). The TOC in the planted group, which fluctuated, increased from 11.54 ± 0.79 g kg^−1^ (Jan) to 15.1 ± 3.62 g kg^−1^ (Nov), and the TOC was lowest in Apr (9.64 ± 2.38 g kg^−1^). The difference in the TOC between the planted and control was greatest in July (4.75 g kg^−1^). The TOC of the control ranged from 8.97 ± 2.00 g kg^−1^ to 10.9 ± 1.76 g kg^−1^, which was significantly lower than that of the planted group (*p* < 0.05) (Figure 2B). The planted TN was also significantly higher than the control (*p* < 0.05), and the highest differences between the planted and control groups were in Jan and July, when the planted TN values were 41.2% and 37.0% higher than those of the control, respectively. The lowest differences occurred in Apr and Sep, when the planted TN values were only 10.8% and 6.72% higher than those of the control, respectively, while the planted TN was lowest in Apr (1.44 ± 0.03 g kg^−1^) and Sep (1.43 ± 0.04 g kg^−^^1^) (Figure 2C). The differences in the AN between the two groups was also highest in Jan (25.3 mg kg^−1^) and July (29.1 mg kg^−1^), the maximum AN values of the planted group were in July (183 ± 15.2 mg kg^−1^) and Nov (179 ± 23.0 mg kg^−1^), and the planted AN was significantly higher than that of the control (*p* < 0.05) (Figure 2D). The planted TP was lowest in Sep (0.249 ± 0.035 g kg^−1^) and highest in July (0.284 ± 0.011 g kg^−1^), while the control TP was lowest in July (0.19 ± 0.02 g kg^−1^). Both the planted AP and control AP were highest in Apr, at 1.97 ± 0.96 mg kg^−1^ and 1.32 ± 0.54 mg kg^−1^, respectively. The lowest AP for the planted group was in Sep (1.53 ± 0.26 mg kg^−1^) and that of the control was in Jan (0.48 ± 0.62 mg kg^−1^). The TP and AP of the planted group were significantly higher than those of the control (*p* < 0.05) (Figure 2E,F). The planted TK ranged from 17.8 ± 4.16 g kg^−1^ (July) to 19.83 ± 4.53 g kg^−1^ (Sep), while that of the control ranged from 10.1 ± 0.41 g kg^−1^ (Jan) to 14.6 ± 2.3 g kg^−1^ (Apr). Both the planted and control groups had the lowest AK in July, at 80.0 ± 7.34 mg kg^−1^ and 55.7 ± 14.6 mg kg^−1^, respectively. The highest AK in the planted group was 97.6 ± 19.4 mg kg^−1^ (Sep), while that of the control was 65.8 ± 1.74 mg kg^−1^ (Apr). The planted group had significantly higher TK and AK than the control (*p* < 0.05) (Figure 2G,H). The results indicated that *Sarcandra glabra* in-forest planting significantly enhanced the soil nutrients and soil carbon storage.

### 3.2. The Variation in the AMF Community

Our results showed that the Shannon index was highest in Jan, at 3.64 ± 0.23 and 3.76 ± 0.13 for the planted and control groups, respectively. The Shannon indexes reduced to 2.62 ± 0.5 (Planted) and 2.14 ± 0.28 (Control) in July. The Shannon index in Sep reached a level near that of Jan for both the planted and control groups, and the Shannon index values in Nov were about level with those in July (Figure 3A). The Chao index showed a similar variation trend to the Shannon index (Figure 3B). No significant differences in the AMF Shannon index and Chao index were found between the planted and control groups (*p* > 0.05), which indicated that the AMF diversity of *Cunninghamia lanceolata* was not significantly altered by *Sarcandra glabra* in-forest planting (Figure 3A,B). We compared the relative abundances (RAs) of the top 20 AMF OTUs. The results showed that the planted and control groups shared nearly the same amount of OTUs, and the RAs of most OTUs showed no significant differences between the planted and control groups, especially in Jan, Apr, and Sep (Figure 3C). Meanwhile, the RAs of a few OTUs showed significant differences between the planted and control groups, mainly in July and Nov. For example, the RA of OTU442 in the control group was about 11 times that of the planted group, and the RA of OTU66 in the planted group was about 11.0%, but it was only 0.07% in the control (Figure 3C). The Pearson analysis found that the RAs of most OTUs were only significantly correlated with one or two nutrients, except for OTU442, OTU231, OTU63, and OTU215 (Figure 3D). Moreover, none of the soil nutrients were significantly correlated with OTU276, OTU64, OTU66, OTU229, OTU47, OTU251, OTU114, or the others (Figure 3D). This indicated that the change in the soil nutrients caused by *Sarcandra glabra* in-forest planting had only limited effects on the RA of AMF OTU. Moreover, the TK was significantly correlated with seven OTUs, while the TN and TP were significantly correlated with four OTUs and one OTU, respectively. This indicated that TK may play a more important role in governing the AMF community of *Cunninghamia lanceolata* (Figure 3D). OTU231, OTU63, OTU221, and OTU215 were significantly correlated with TOC, which indicated its possible function of releasing or generating soil organic carbon (Figure 3D).

As the *p*-values of the ANOISM tests were all more than 0.001, this indicates that the composition of the AMF community over the whole year or at any sampling time showed no significant differences between the planted and control groups (Figure 4). Moreover, among all the sampling times, July had the lowest *p*-value (0.10), which indicated that the compositional differences between the planted and control groups were relatively higher in July than at the other sampling times (Figure 4D). This result was congruent to the previous result, indicating that certain OTUs showed significant differences in the RA between the planted and control groups in July. The PCoA analysis found that *Sarcandra glabra* cultivation led to insignificant changes in the composition of the *Cunninghamia lanceolata* AMF community.

### 3.3. The Economic Analysis of Sarcandra Glabra In-Forest Planting

The price of *Sarcandra glabra* seeds (reproductive branches) was about each CNY 0.4 and the cost of the seeds (reproductive branch) in first year was about CNY 12,000 hm^−2^. The labor costs in first year amounted to about CNY 13,500 h^−2^, while in the following years they amounted to about CNY 3200 hm^−2^, because the soil preparation and the removal of grasses in the first year require a larger labor force (Figure 5). Similarly, the cost of fertilizer of first year was about CNY 4000 hm^−2^, while in the other years it was about CNY 3600 hm^−2^ (Figure 5). The first harvest of *Sarcandra glabra* leaves took place in the third year, and the yield was about 1500 kg (dry mass) hm^−2^. According to the market price of CNY 2.4 kg^−1^, the income from the first harvest year was as high as CNY 54,000 hm^−2^ (Figure 5). And the following years, the estimated yield was about 1000 kg hm^−2^, and the income was about CNY 36,000 hm^−2^ (Figure 5). The *Sarcandra glabra* can generally be harvested after 7–10 years [13,14]. The total income ranged from CNY 127,700 hm^−2^ (7 years cultivation) to CNY 215,300 hm^−2^ (10 years cultivation), and the average annual income ranged from CNY 18,242 hm^−2^ y^−1^ to CNY 21,530 hm^−2^ y^−1^ (Figure 5).

## 4. Discussion

### 4.1. The Influences of Sarcandra Glabra Cultivation on Soil Nutrients

In this research, we found that *Sarcandra glabra* cultivation slightly, but significantly, lowered the soil pH (about 0.1), which was potentially due to the application of inorganic fertilizers. The roots release hydrogen ions into the soil, where they exchange and assimilate nutrients such as potassium ions and ammonium ions [27]. We found a weakly increasing trend of the TOC in the planted group, and the planted TOC was significantly higher than that of the control, which indicated that *Sarcandra glabra* cultivation accelerated the topsoil TOC accumulation. We suggest that this was due to the input of additional nitrogen and phosphorus during cultivation. The C:N ratio of our sampling sites ranged from 7.36–10.5, and the C:P ratio ranged from 78–166, which indicated a relative carbon-limited state, according to Redfield stoichiometry [28]. In nitrogen- and phosphorus-limited soils, the nitrogen and phosphorus additions tend to stimulate microbial respiration and the utilization of recalcitrant organic carbon, which reduces the soil’s organic carbon storage [28,29]. Meanwhile, in carbon-limited soils, the nitrogen and phosphorus additions tend to stimulate soil microbial activity and accelerate plant litter decomposition and soil organic carbon sequestration, which enhances the soil organic carbon storage [28,30]. Thus, our results suggest that, for relatively carbon-limited red soil regions, scientific fertilization during herb cultivation would benefit the process of topsoil organic carbon storage. We found that, in July, Nov, and Jan, the TN differences between the planted and control groups were magnified by the fertilizations at the end of Apr and the end of Sep (Figure 2C). Moreover, the TN differences were relatively lower at the beginning of Apr and Sep, just before the fertilization. In addition, the AN, TP, and AP showed a similar annual variation (Figure 2D–F). These results indicated that the applied nitrogen and phosphorus were efficiently assimilated by *Sarcandra glabra*, which reduced the soil nitrogen differences and phosphorus differences between the fertilized planted soil and unfertilized control soil. Apr and Sep were, respectively, the key stages of nutritional growth and reproductive growth, and the *Sarcandra glabra* demand for nutrients were higher than in the other stages [14]. Our results indirectly confirmed that appropriate fertilization in these two stages can promote the growth and production of *Sarcandra glabra*. Moreover, the results showed that the TP of the control was lowest in July (Figure 2E), which indicated a quick phosphorus consumption by the plant community during spring, which lowered the soil TP as TP was transported to the up-ground communities. Meanwhile, during autumn and winter, the TP was transported back to the soil through litters, which led the soil TP to increase again [31,32,33]. These results indicated the potential competition for phosphorus between *Cunninghamia lanceolata* and *Sarcandra glabra* during spring and highlighted the importance of spring phosphorus fertilization for *Sarcandra glabra* cultivation. Both the planted and control groups had the highest AP in Apr in this research (Figure 2E,F). Apr is rainy season, and the high soil moisture stimulates soil microbial activity, which favors the mineralization of organic phosphorus, as well as the dissolution of rocky phosphorus [33]. The released inorganic phosphorus supported the quick development of the plants in Apr [33], which caused the lower soil TP at the end of rainy season (July). Similarly, the control TK and AK were also highest in Apr (Figure 2G,H), which indicated that the plant community had a high demand for potassium in spring [34]. Above all, our results suggest that *Sarcandra glabra* cultivation significantly enhanced the soil nitrogen, phosphorus, and potassium contents and stimulated topsoil organic carbon storage, which had positive effects on the needle forest ecosystems. However, it has also been reported that AMF communities are sensitive to agricultural practices, as they play an important role in helping the hosts to assimilate nitrogen and phosphorus, and conventional agriculture systems exert higher negative impacts on the AMF spore pool [35]. Thus, we still need to combine ecosystem phenology and the key stage of *Sarcandra glabra* development in order to conduct scientific fertilization, reduce the potential competition for nutrients between *Cunninghamia lanceolata* and *Sarcandra glabra*, and ensure forest health and *Sarcandra glabra* production.

### 4.2. The Influences of Sarcandra Glabra Cultivation on the AMF of Cunninghamia lanceolata

Our results showed that the α diversity of AMF was not reduced by *Sarcandra glabra* cultivation (Figure 3A,B). A community with a higher α diversity tends to possess higher functional redundancy and higher functional resistance to environmental stress [36], and the cooperation between diverse community members also accelerates efficient ecological processes [37]. The AMF plays important roles in several ecological processes, including the host plant nutrient uptake [17], the regulation of the plant community dominance, diversity, and primary production [38], and the generation of recalcitrant soil organic carbon [16], and the reduction of AMF α diversity may cause potential damage to the forest’s health. Several studies have reported that certain types of microbes were selected and enriched due to fertilization, which reduced the soil microbial community diversity [39,40]. It was also reported that AMF α diversity was seriously reduced by conventional fertilization management rather than organic management [10]. The AMF links the soil to the plant roots, and the α diversity of an AMF is governed by both sides [16]. Our results indicated that the AMF of *Cunninghamia lanceolata* was mainly governed by the host state instead of the soil nutrients, which was further supported by the fact that most OTUs were insensitive to nutrients, as their RAs were not significantly correlated with the nutrients. *Sarcandra glabra* cultivation led to insignificant changes in the RA for most OTUs, and only few OTUs showed significant differences between the planted and control groups, mainly in July. As previously discussed, the plants have a much higher demand for nutrients during the spring growth season, which may lead to potential competition for nutrients, and force spruce slightly adjust the AMF community composition according to its nutritional state [17]. Thus, we suggest monitoring the further improvement of fertilization management so as to ensure the sufficient nutrient uptake by forest keystone plants. Surprisingly, we found that the RAs of seven OTUs were significantly correlated with th TK, while only one was significantly correlated with the TP. It has been widely reported that the AMF accelerates the nitrogen and phosphorus acquisition of terrestrial plants [18], while the potassium acquisition is relatively less frequently reported [41]. Though potassium is one of the most abundant elements in soil, its biological availability is limited by the plant cell machinery, which is reportedly closely connected with arbuscular mycorrhizal and ectomycorrhizal symbiosis [41]. Several studies have reported that AMF accelerated the potassium enrichment in *Zea mays* roots teles [42], in *Pelargonium peltatum* shoots [43], and in *Lactuca sativa* leaves [44]. As potassium has significant effects on variated physiological processes of spruce, such as transpiration and leaf diffusive resistance [34], our results indirectly highlighted the possible functions of AMF in *Cunninghamia lanceolata* potassium acquisition. In this research, OTU442, OTU234, and OTU63 had relatively higher RAs in their communities, and their RAs were significantly correlated with several nutrients rather than only one nutrient, which indicated that they played potentially important roles in helping *Cunninghamia lanceolata* to acquire those nutrients. They all belong to the family *Glomeraceae* and the genus *Glomus*. In Glomus species, anastomoses were observed both within and between the mycelial networks of the same isolates, which could enhance the sink-regulated redistribution of resources within a single fungal colony and could increase the nutrient transport efficiency [45]. This is a potential explanation for the fact that these OTUs act as nutrient transport centers in the AMF community. We found that several OTUs were significantly positively related to the TOC (Figure 3D). The AMF receives organic carbon from host plants, releases organic carbon into the soil, and create fungal senescence, which contains a recalcitrant glycoprotein that can take years to decades to degrade in the soil [46]. Moreover, these OTUs may play key roles in soil organic carbon storage. Above all, our results showed that *Sarcandra glabra* cultivation only led to insignificant differences in the diversity and composition of the *Cunninghamia lanceolata* AMF, while we need to improve current fertilization measurements in order to ensure the minor differences, which will not be magnified in the near future, potentially damaging the ecosystem sustainability.

### 4.3. The Income from Sarcandra Glabra In-Forest Planting

Since the ecologically friendly economy development theory, asserting that “clear waters and green mountains are as good as mountains of gold and silver”, was proposed in 2005 [47,48], agroforestry in China, especially in southern China, has undergone unprecedented development, which set an example for the construction of ecological civilizations and the harmonic coexistence of human and nature for the whole world [47,48]. Taking Jiangxi Province as an example, the total production value of the in-forest economy exceeds 170 billion [49], with over 2.6 million hectares of forest being engaged in production activities, and more than 3 million people employed as labors, among whom about 12% were defined as being in extreme poverty [49]. The in-forest planting of herbs enhanced the income of 6.1 thousand poor people by about CNY 15,000 per family within a year [49,50]. In our research, the average annual income from *Sarcandra glabra* cultivation ranged from CNY 18,242 hm^−2^ to CNY 21,530 hm^−2^ (Figure 5), which is about 20–40% higher than the average income from herb cultivation, which indicated the increase in the market demands for *Sarcandra glabra*. More importantly, our results confirmed that *Sarcandra glabra* in-forest cultivation did not necessarily impact the forest negatively, and it might even offer positive ecological effects in accelerating soil carbon storage and mitigating climate change (Figure 2B) [4]. In-forest planting forestry is reported to yield a higher economic income than pure timber-oriented forestry [1,3]. Our results showed that *Sarcandra glabra* in-forest cultivation offered an additional CNY 10,900 hm^−2^ in the first 3 years of cultivation and CNY 127,700 hm^−2^ after 7 years of cultivation. With the improvement of the agronomic measurements and the following elongation of the cultivation periods, the total additional income could be as high as CNY 215,300 hm^−2^ after 10 years. A large-diameter, timber-oriented agroforestry system always lasts for 30~50 years, and *Sarcandra glabra* in-forest cultivation during suitable periods constitutes an efficient utilization of these long-lasting forest resources, which offers considerable additional income to foresters and scientific confirmation of the policy that “clear waters and green mountains are as good as mountains of gold and silver” [47,48]. Here, we call for further research focusing on relative agronomic improvements in order to enhance yields and reduce negative ecological effects in the long term.

## 5. Conclusions

In this research, we sampled soil and AMFs from a 3-year-old *Sarcandra glabra* planted forest and a control forest and compared the annual variation in the soil nutrients and the AMF community between the two forests. We found that *Sarcandra glabra* in-forest planting significantly enhanced the soil topsoil carbon storage, while leading to insignificant variations in the diversity and composition of the *Cunninghamia lanceolata* AMF community, which indicated that *Sarcandra glabra* in-forest planting was beneficial for the forest and was ecologically safe for the *Cunninghamia lanceolata* AMF. Meanwhile, at the same time, *Sarcandra glabra* in-forest planting offered considerable incomes to foresters, which is a clear example of the “clear waters and green mountains are as good as mountains of gold and silver” policy. However, we need to further improve the fertilization management techniques in order to reduce the competition for nutrients between *Sarcandra glabra* and *Cunninghamia lanceolata* and to ensure production, as well as forest health and sustainability.

## Figures and Tables

**Figure 1 microorganisms-10-01844-f001:**
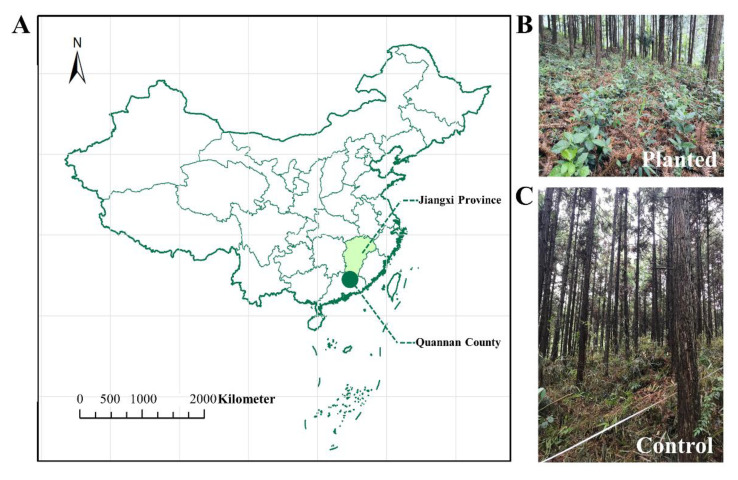
The sampling site. (**A**) depicts the detailed location of the site, and (**B**,**C**) depict the vegetations of the planted and control groups, respectively.

**Figure 2 microorganisms-10-01844-f002:**
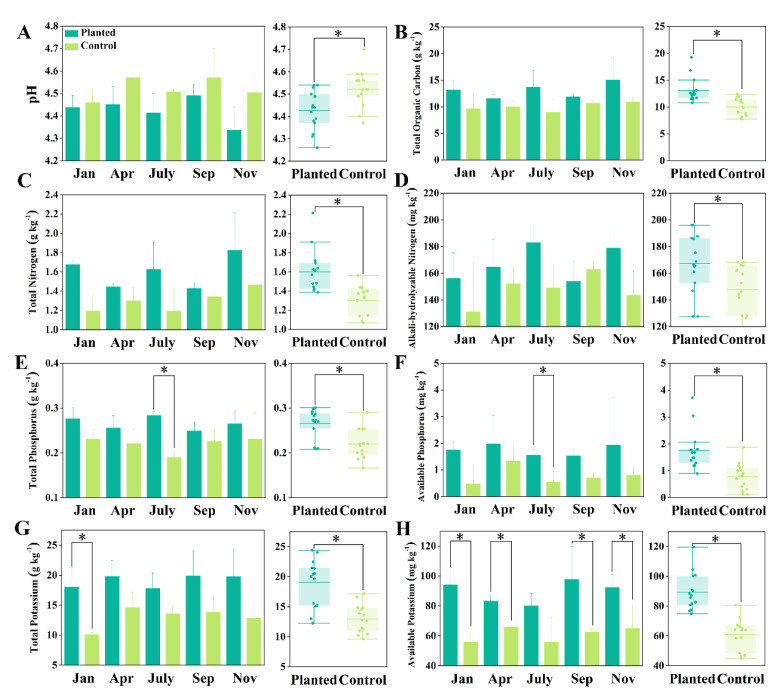
The annual variation in the soil pH and basic nutrients. The star marker indicates significant differences between the two groups (one-tail *T*-test, *p* < 0.05), and the error bar indicates the standard deviation (*n* = 3 in the left part and *n* = 15 in the right part of subfigure (**A**–**H**)).

**Figure 3 microorganisms-10-01844-f003:**
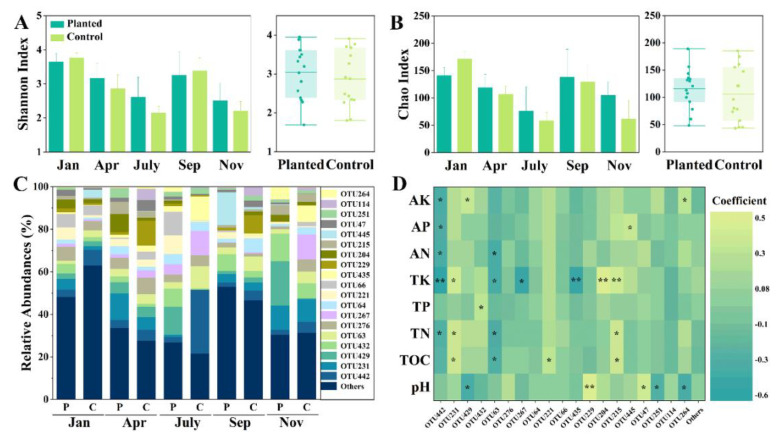
The annual variation in the basic AMF community properties. The error bar in (**A**,**B**) indicates the standard deviation (*n* = 3 in the left part and *n* = 15 in the right part of each subfigure). P and C in the *x*-axis of (**C**) are abbreviations of planted and control, respectively. In (**D**), the star marker and double-star marker indicate significantly correlations at *p* < 0.05 and *p* < 0.01 level, respectively.

**Figure 4 microorganisms-10-01844-f004:**
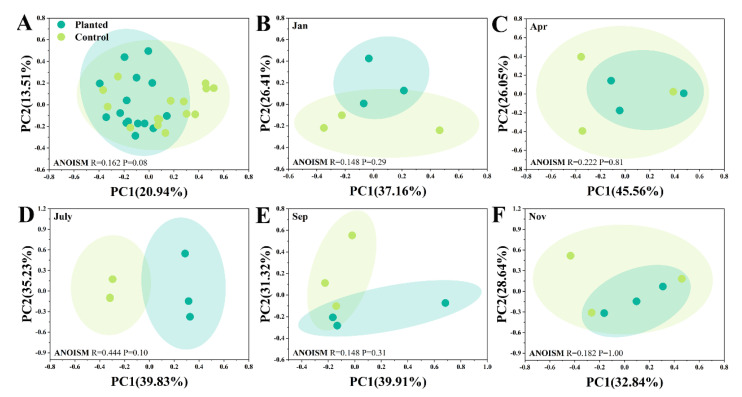
The PCoA analysis of the AMF community’s composition on the OTU level: (**A**) depicts annual differences between the planted and control groups, while (**B**–**F**) depict each sampling time. A *p* < 0.05 indicates a significant difference between the two groups.

**Figure 5 microorganisms-10-01844-f005:**
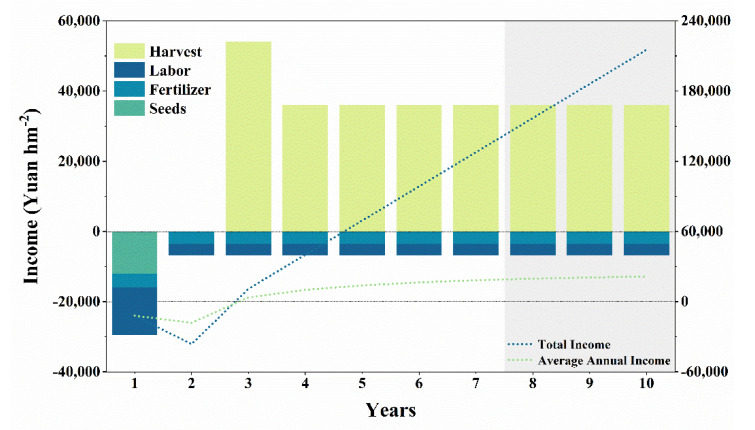
The costs of, and income from, in-forest *Sarcandra glabra* planting. The left *y* axis indicates the costs of, and income from, the main parts, while the right *y* axis indicates the variation in the total income and average annual income. As *Sarcandra glabra* planting generally lasts for 7–10 years, the shaded area indicates potential cost and income that varies with planting age.

## Data Availability

The raw sequences were uploaded to NCBI, and the project number is PRJNA862268.

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
