# Peer review of "In-Forest Planting of High-Value Herb Sarcandra glabra Enhances Soil Carbon Storage without Affecting the Diversity of the Arbuscular Mycorrhiza Fungal Community and Composition of Cunninghamia lanceolata"

_microorganisms, 2022, doi:10.3390/microorganisms10091844_

Round 1

Reviewer 1 Report

Comments on the paper entitled „In-forest planting of high value herb Sarcandra glabra enhances  soil carbon storage without affecting the arbuscular mycorrhiza  fungal community diversity and composition of Cunninghamia  lanceolata” Manuscript ID: microorganisms-1902482

General comments

The study concerns the assessment of the impact of Sarcandra glabrana cultivation in needle forest, where the dominant species is Cunninghamia lanceolata, on changes in the content of nutrients important from the fertilization point of view and the community of AMF. The economic aspect of such cultivation was also assessed.

The work requires minor corrections before it will be published.

Specific comments

1.     Line 114: How many repetitions were measured?

2.     Line 115: Typically soil pH is measured in 1 M KCl. Please consider this for the future.

3.     Line 116: Please delete the word "were"

4.     Line 177 and 181: The authors describe the Shannon index and Chao index but did not provide any information on this in the Material and methodology section. Please complete it, among others write what the individual indexes are for.

5.     Line 191: Pearson analysis was not mentioned in the methodology. Please complete it.

6.     Line 245 and 260: And what is the contribution of AMF to nitrogen and phosphorus uptake? After all, AMF takes up N and P, especially phosphorus, and passes it on to the plant.

About Arbuscular Mycorrhizal Fungi can be found in Pathogens 2022, 11, 844. https://doi.org/10.3390/ pathogens11080844

7.     Line 280: Is only Sarcandra glabra cultivation significantly enhanced soil nitrogen, phosphorus and potassium content? Does this plant release such compounds? Can fertilization affect the content of N and P in the soil?

8.     Has AMF been tested on Sarcandra glabra roots?

Author Response

General comments

The study concerns the assessment of the impact of Sarcandra glabrana cultivation in needle forest, where the dominant species is Cunninghamia lanceolata, on changes in the content of nutrients important from the fertilization point of view and the community of AMF. The economic aspect of such cultivation was also assessed.

The work requires minor corrections before it will be published.

Specific comments

  1. Line 114: How many repetitions were measured?

Response:Line 124 “Three repetitions were measured for soil properties of each sample, which was corre-sponded to one molecular sequencing of that sample.”

  1. Line 115: Typically soil pH is measured in 1 M KCl. Please consider this for the future.

Response:Thanks, we’ll try that method in the future.

  1. Line 116: Please delete the word "were"

Response: Deleted

  1. Line 177 and 181: The authors describe the Shannon index and Chao index but did not provide any information on this in the Material and methodology section. Please complete it, among others write what the individual indexes are for.

Response: Line 137~141 “The Shannon index and Chao index were calculated in online platform (http://www.magichand.online) according to the reads number of each OTUs in every sample. Higher Chao index indicates that a sample contains higher none-zero reads OTUs. Higher Shannon index indicates that a sample contains more diversified OTU composition, not only more none-zero reads but also more evenly distributed reads [24]”

  1. Line 191: Pearson analysis was not mentioned in the methodology. Please complete it.

Response: Line 147 “the pearson analysis was conducted in SPSS version 26.”

  1. Line 245 and 260: And what is the contribution of AMF to nitrogen and phosphorus uptake? After all, AMF takes up N and P, especially phosphorus, and passes it on to the plant.

About Arbuscular Mycorrhizal Fungi can be found in Pathogens 2022, 11, 844. https://doi.org/10.3390/ pathogens11080844

Response: Line 288-Line 291 “However, AMF were also reported sensitive to agricultural practices as it plays an im-portant role on help host assimilating nitrogen and phosphorus, and conventional agri-culture systems exerted higher negative impacts on AMF spore pool [34]”, and thanks for the suggestion on the key references.

  1. Line 280: Is only Sarcandra glabra cultivation significantly enhanced soil nitrogen, phosphorus and potassium content? Does this plant release such compounds? Can fertilization affect the content of N and P in the soil?

Response: Yes, the herb cultivation might influence soil microbial community via not only fertilization but also other ways, including releasing secondary metabolites into soil. We are not sure the quantity of secondary metabolites accumulation during Sarcandra glabra cultivation, and can not determine its potential effects on soil microbial community yet, especially in the long run. The TN and TP were mainly enhanced by fertilization, though most of them were assimilated by Sarcandra glabra according to our results, that’s why the microbial community didn’t change significantly.

  1. Has AMF been tested on Sarcandra glabraroots?

Response: No, we mainly focused on the influence of Sarcandra glabra cultivation on AMF of forest keystone species.

Lastly, we sincerely thanks for the kind suggestions and corrections, they really helped improving the quality of this manuscript. Best wishes to the reviewers.

Reviewer 2 Report

Article: Manuscript ID: microorganisms-1902482

Title: In-forest planting of high value herb Sarcandra glabra enhances soil carbon storage without affecting the arbuscular mycorrhiza fungal community diversity and composition of Cunninghamia lanceolata

Hanchang Zhou, Tianlin Ouyang, Liting Liu, Shiqi, Xia, Quanquan Ji

The manuscript deals with the cultivation of Sarcandra glabra, an interesting herb whose cultivation increases the income of foresters. The manuscript concerns research that aims to verify the effect of cultivating Sarcandra glabra, which requires fertilisation twice a year, in a 25-year-old Cunninghamia lanceolata plantation. In particular, the authors studied the soil nutrients (pH, total organic carbon, total nitrogen, total phosphorus, total potassium, alkali-hydrolyzable nitrogen, available phosphorus, available potassium) and the community of arbuscular mycorrhizal fungi in the forest trees in the cultivated area compared to the forest control. They monitored both nutrients and the community of arbuscular mycorrhizal fungi in forest and in-forest planting at January, April, July, September and November. Furthermore, they provided an economic analysis of this crop that offers significant additional income to foresters.

The authors demonstrated that planting Sarcandra glabra in forests significantly increased carbon storage in the topsoil. No significant changes were found in the diversity and composition of the arbuscular mycorrhizal fungi community of Cunninghamia lanceolata, despite fertilisation applied twice a year, at least three years after planting.

The comparison between the area planted with Sarcandra glabra and the control indicates that the planting of Sarcandra glabra in the forest did not disturb the Cunninghamia lanceolata forest in terms of diversity and composition of the arbuscular mycorrhiza fungal community.

The work is interesting because monitored the soil nutrient coupled with the community of arbuscular mycorrhizal fungi. The weakness of the work is the short period examined.

The abstract is well done, briefly presenting the background, highlights of the methodology, results and conclusions. I suggest avoiding the word used in the title and adopting an alphabetic order for the keywords. 

The introduction is well structured, and the reference list cover the relevant literature adequately. Some terms are unusual and I suggest to check them.

The methodological section is adequate, but the experimental design could be better explained. A brief description of cultivation of this herb should be provided as the reference is in Chinese. The methods are documented, and the figure is useful. Sub-Figure 1A should be improved. More details on the sampling site and on sampling fine roots would be desirable. The paragraph Statistics needs improvement as a numbers of tests were used but not indicated here.

The results, discussion and conclusion sections are well structured. Figures are useful for understanding the proposed data. Greater accuracy must be placed in the drafting of the captions because the figures must be self-explanatory to help the reader understand results.

The language is understandable, even if some revision is needed, but I don't feel like judging.

The manuscript is rigorous. It has a significance to the field both for research and practical applications, and it has an interest to the general audience of “Microorganisms”.

List of observation

line 2 Please, check the font

line 34 Consider avoiding replication of the words in title. It is preferable an alphabetic order in keywords, but it is not mandatory.

line 67 I am not native English language, but I do not remember “needle forest” in scientific articles. Furthermore, “in where” is incorrect. Please, check.

line 90 It should be “average diameter at breast height”

lines 92-93 The patronymic of the latin name of the species is not in italics. Check the document, please.

line 93 “2017.02” is not incorrect but February 2017 is more friendly.

lines94-96 A brief description of cultivation of this herb should be provided as the reference is in Chinese.

Figure 1 A is not clear, use a better image.

line 104 How did you selected the sub samples?

line 100 Avoid abbreviation

line 111 “about 50 fine roots” The sampling method of fine roots of the Cunninghamia is not explained and the reference seems not adequate. Please revise.

line 137-142 This paragraph needs improvement: you used a numbers of tests indicated in results, so please indicate them and the dataset used.

lines 146, 148, 152, 157, It is a formal observation: please, control the instruction to authors.

line 174 The figure should be self-explanatory. Indicate the meaning of the bars, in caption.

line 203 The figure should be self-explanatory. Please check A and B: the bars represent….

line 207 ANOSIM the acronym should be made explicit and briefly explained in 2.5 Statistics.

lines 217-218 The figure should be self-explanatory. Please check: D, E, F?

line 237 It is not clear the meaning of the sentence: “The shaded area indicates possible cost and income”

Author Response

Article: Manuscript ID: microorganisms-1902482

Title: In-forest planting of high value herb Sarcandra glabra enhances soil carbon storage without affecting the arbuscular mycorrhiza fungal community diversity and composition of Cunninghamia lanceolata

Hanchang Zhou, Tianlin Ouyang, Liting Liu, Shiqi, Xia, Quanquan Ji

The manuscript deals with the cultivation of Sarcandra glabra, an interesting herb whose cultivation increases the income of foresters. The manuscript concerns research that aims to verify the effect of cultivating Sarcandra glabra, which requires fertilisation twice a year, in a 25-year-old Cunninghamia lanceolata plantation. In particular, the authors studied the soil nutrients (pH, total organic carbon, total nitrogen, total phosphorus, total potassium, alkali-hydrolyzable nitrogen, available phosphorus, available potassium) and the community of arbuscular mycorrhizal fungi in the forest trees in the cultivated area compared to the forest control. They monitored both nutrients and the community of arbuscular mycorrhizal fungi in forest and in-forest planting at January, April, July, September and November. Furthermore, they provided an economic analysis of this crop that offers significant additional income to foresters.

The authors demonstrated that planting Sarcandra glabra in forests significantly increased carbon storage in the topsoil. No significant changes were found in the diversity and composition of the arbuscular mycorrhizal fungi community of Cunninghamia lanceolata, despite fertilisation applied twice a year, at least three years after planting.

The comparison between the area planted with Sarcandra glabra and the control indicates that the planting of Sarcandra glabra in the forest did not disturb the Cunninghamia lanceolata forest in terms of diversity and composition of the arbuscular mycorrhiza fungal community.

The work is interesting because monitored the soil nutrient coupled with the community of arbuscular mycorrhizal fungi. The weakness of the work is the short period examined.

The abstract is well done, briefly presenting the background, highlights of the methodology, results and conclusions. I suggest avoiding the word used in the title and adopting an alphabetic order for the keywords. 

The introduction is well structured, and the reference list cover the relevant literature adequately. Some terms are unusual and I suggest to check them.

The methodological section is adequate, but the experimental design could be better explained. A brief description of cultivation of this herb should be provided as the reference is in Chinese. The methods are documented, and the figure is useful. Sub-Figure 1A should be improved. More details on the sampling site and on sampling fine roots would be desirable. The paragraph Statistics needs improvement as a numbers of tests were used but not indicated here.

The results, discussion and conclusion sections are well structured. Figures are useful for understanding the proposed data. Greater accuracy must be placed in the drafting of the captions because the figures must be self-explanatory to help the reader understand results.

The language is understandable, even if some revision is needed, but I don't feel like judging.

The manuscript is rigorous. It has a significance to the field both for research and practical applications, and it has an interest to the general audience of “Microorganisms”.

List of observation

line 2 Please, check the font

Response: We checked the font according to MDPI LaTeX templates

line 34 Consider avoiding replication of the words in title. It is preferable an alphabetic order in keywords, but it is not mandatory.

Response: We changed keywords in alphabetic order.

line 67 I am not native English language, but I do not remember “needle forest” in scientific articles. Furthermore, “in where” is incorrect. Please, check.

Response: We use “coniferous forest” instead and deleted the “in”.

line 90 It should be “average diameter at breast height”

Response: Thanks, we had corrected it.

lines 92-93 The patronymic of the latin name of the species is not in italics. Check the document, please.

Response: Thanks, we corrected them and check the whole manuscript.

line 93 “2017.02” is not incorrect but February 2017 is more friendly.

Response: Thanks, we used “February 2017” instead.

lines94-96 A brief description of cultivation of this herb should be provided as the reference is in Chinese.

Response: We added “A 10~15cm branches of Sarcandra glabra was used for vegetative reproduction. The bottom of branches were rinsed to solution of ABT3 for 2~3min, then the branches were buried into 10cm deep soils every 5~10cm intervals.” accordingly in Line 94-96.

Figure 1 A is not clear, use a better image.

Response: We changed a clearer one, and uploaded the figure in TIF further.

line 104 How did you selected the sub samples?

Response: Line 104 “Three subsample plots (20m×20m) were set randomly in both Planted and Control with each one was at least 20m away from another.”

line 100 Avoid abbreviation

Response: Changed “Km” to “Kilometer”

line 111 “about 50 fine roots” The sampling method of fine roots of the Cunninghamia is not explained and the reference seems not adequate. Please revise.

Response: We rewrote Line112~line120 “Root samples of the Cunninghamia lanceolata were dug out by shovel from surface soils (0–20 cm) at 10 randomly selected locations in each subsample plot. Each of the 10 sampling locations was located at least 2 m apart. we then used these roots and selected the distal first and second branch orders of live Cunninghamia lanceolata roots, considered as absorp-tive roots [19], for further analyses. Identification of the live roots of the Cunninghamia lan-ceolata was based on root color, morphology and elasticity of their tissues. Following col-lection, all absorptive root samples were gently washed free from soil by tap water and then deionized water. We selected ca. 50 absorptive root segments for DNA extraction[20].”

line 137-142 This paragraph needs improvement: you used a numbers of tests indicated in results, so please indicate them and the dataset used.

Response: We changed Line 154-158 into “ANOISM-test were used to define the significance of differences between Control and Planted, and the P less than 0.05 indicated significance differences. The significant differ-ence of basic soil properties and diversity indices between Planted and Control were checked using t-test in SPSS version 26, the P less than 0.05 indicated significance differ-ences. And the pearson analysis was conducted in SPSS version 26, the P less than 0.05 indicated significantly correlated.”

lines 146, 148, 152, 157, It is a formal observation: please, control the instruction to authors.

Response: We deleted the postulation description in this result part.

line 174 The figure should be self-explanatory. Indicate the meaning of the bars, in caption.

Response: We added “the error bar indicates standard deviation (n=3 in left part and n=15 in right part of each subfigure)”

line 203 The figure should be self-explanatory. Please check A and B: the bars represent.

Response: We added “The error bar in subfigure-A and subfigure-B indicates standard deviation (n=3 in left part and n=15 in right part of each subfigure).”

line 207 ANOSIM the acronym should be made explicit and briefly explained in 2.5 Statistics.

Response: We added “ANOISM (Analysis of similarities)-test were used to define the significance of differences between Control and Planted, and the P less than 0.05 indicated significance differences.” in 2.5

lines 217-218 The figure should be self-explanatory. Please check: D, E, F?

Response: We rewrote the Caption into “Figure 4. The PCoA analysis of AMF community composition on OTU level, subfigure-A depicts annual differences between Planted and Control while B~C depict each sampling time, the P<0.05 indicate significant difference between two groups.”. And we checked subfigure D, E and F again.

line 237 It is not clear the meaning of the sentence: “The shaded area indicates possible cost and income”

Response: We rewrote as “As the Sarcandra glabra planting generally lasts 7~10 years, the shaded area indicates potential cost and income that varies with planting age.”

Lastly, we thank for these kind corrections and suggestions mentioned by dear reviewer2, them really helped improved our manuscript quality, and taught me a lot about how to write a good paper. We sincerely wish you all better.
